# Psychobiotics in Depression: Sources, Metabolites, and Treatment—A Systematic Review

**DOI:** 10.3390/nu17132139

**Published:** 2025-06-27

**Authors:** Angelika Śliwka, Magdalena Polak-Berecka, Kinga Zdybel, Agnieszka Zelek-Molik, Adam Waśko

**Affiliations:** 1Department of Biotechnology, Microbiology and Human Nutrition, Faculty of Food Science and Biotechnology, University of Life Sciences in Lublin, Skromna 8, 20-704 Lublin, Poland; magdalena.polak-berecka@up.lublin.pl (M.P.-B.); kinga.zdybel@up.lublin.pl (K.Z.); adam.wasko@up.lublin.pl (A.W.); 2Department of Brain Biochemistry, Maj Institute of Pharmacology, Polish Academy of Sciences, Smetna 12, 31-343 Krakow, Poland; zelek@if-pan.krakow.pl

**Keywords:** psychobiotics, microbiota, gut–brain axis, mental health, depression, HPA axis, systematic review

## Abstract

**Background**: Depression and other stress-related mental disorders are the leading causes of disability worldwide, making them a significant global health challenge. This systematic review aimed to determine the effects of psychobiotic microorganisms on mental health outcomes, with particular focus on their sources, metabolites, and therapeutic potential for depression. **Methods**: A systematic review following PRISMA guidelines was conducted using publications from 2020 to 2024 in Web of Science, Scopus, and PubMed databases. Inclusion criteria encompassed studies examining psychobiotics and their effects on mental health in humans and experimental animals. Risk of bias assessment was performed using the Cochrane Risk of Bias Tool (ROB 2). **Results:** Of 369 identified articles, 45 met inclusion criteria. The predominant psychobiotic strains belonged to *Lactobacillus* (45.5%) and *Bifidobacterium* (29%) genera. Strain sources included commercial preparations (24%), human-derived (16%), and food-derived (16%) strains. Psychobiotic bacterial strains produce neuromodulatory metabolites, such as short-chain fatty acids (SCFAs), neurotransmitters (e.g., GABA and serotonin), and indole derivatives that influence the gut–brain axis. Their mechanisms of action include neurotransmitter regulation (27.1%), modulation of the gut microbiota (27.1%), SCFA production (16.9%), and control of inflammatory responses (15.3%). *Lactobacillus plantarum*, *Bifidobacterium breve*, and *Akkermansia muciniphila* demonstrated particularly promising effects. **Conclusions**: Psychobiotics show significant potential as adjunctive and therapeutic agents in depressive disorders through modulation of the gut–brain axis.

## 1. Introduction

Depression is a mental disorder characterized by prolonged periods of low mood and anhedonia, as well as sleep and psychomotor disturbances, changes in appetite or weight, fatigue, or loss of energy [1,2]. Recent 2023 data indicates that approximately 280 million people are affected by depression worldwide, making it the most prevalent psychiatric condition and the leading cause of disability globally [3]. Anxiety and cognitive disturbances often precede or coexist with depression [4,5]. Moreover, depression is commonly observed in the early stages of neurodegenerative diseases [6].

Clinical and preclinical studies suggest that a shared underlying factor in these mental illnesses is exposure to long-term, maladaptive stress [7,8,9,10].

In addition to psychiatric symptoms, chronic stress and elevated glucocorticoid levels have been shown to increase intestinal permeability, a condition known as leaky gut syndrome. This allows gut bacteria and their metabolites—such as short-chain fatty acids, neurotransmitters, and cytokines—to translocate into the bloodstream, leading to disruptions in the composition and function of the gut microbiome, a state referred to as dysbiosis. Recent studies indicate that dysbiosis is an important factor associated with depression and that the effectiveness of antidepressant treatment is closely related to the restoration of a healthy gut microbiota [2,11]. Although several antidepressant strategies are currently available—including pharmacotherapy, psychotherapy, electroconvulsive therapy, and transcranial stimulation—their clinical utility is limited due to undesirable side effects, and more than 35% of patients are resistant to treatment [2]. This underscores the urgent need to identify compounds with greater therapeutic efficacy than those currently available.

Psychobiotics are live microorganisms with potential mental health benefits, modulating the microbiota–gut–brain axis via immune, humoral, neural, and metabolic pathways [12,13]. Among the most frequently studied psychobiotic bacteria are genera such as *Lactobacillus*, *Lactococcus*, *Bifidobacterium*, *Streptococcus*, and *Enterococcus,* which influence this axis through the production of short-chain fatty acids (SCFAs), neurotransmitters, and other bioactive metabolites [13,14]. Notably, SCFAs-producing bacteria, including *Lactobacillus*, *Bifidobacterium*, and *Clostridium*, have been implicated in various psychiatric disorders, highlighting their potential as novel psychobiotics [12].

The gut microbiota communicates with the brain through the gut–brain axis, and psychobiotics can modulate this interaction by synthesizing neuroactive compounds, regulating the hypothalamic–pituitary–adrenal (HPA) axis, and modulating immune responses [14,15]. Their effects include antidepressant and anxiolytic properties, which have been observed in both preclinical and clinical studies. Furthermore, the concept of psychobiotics has been expanded to include inactivated microorganisms with similar mental health benefits, demonstrating positive effects on behavior and microbiota composition even in healthy individuals [16]. Inactivated microbial cells, also known as postbiotics, include metabolites, inactivated cells, and other molecules that support the development of psychobiotic strains in the gut. The use of inactivated microorganisms has several advantages over live organisms, including the lack of risk of infection in susceptible individuals and ease of use in terms of storage and administration [17].

Psychobiotic microorganisms can be found in fermented foods such as yogurt, sauerkraut, and kimchi [18,19]. Healthy dietary patterns rich in pro- and prebiotics play a crucial role in mood regulation through their impact on the gut microbiome [20,21].

Human-derived probiotics are beneficial microorganisms isolated from the gut, vagina, and feces microbiota, rich in *Lactobacillus* and *Bifidobacterium* species [22,23,24]. These strains are characterized by key functional properties, such as survival in an acidic environment and adhesion to intestinal cells, which contribute to health benefits for the digestive system and psychological well-being. However, the use of human-derived probiotics requires rigorous safety assessments, including obtaining the GRAS (Generally Recognized As Safe) status, which guarantees safe therapeutic and supplementary use [25].

Clinical and preclinical studies have demonstrated the potential of psychobiotics in ameliorating conditions such as anxiety, depression, and stress-related disorders [16]. However, it is important to note that the exact mechanisms of action and the specific roles of these microorganisms in modulating the microbiota–gut–brain axis are still not fully understood [13,15]. Further research is essential to unravel the complex interactions between these microorganisms and the central nervous system, paving the way for innovative psychobiotic-based therapies [16,26].

Despite the fact that pharmacological methods of treating depression using psychotropic drugs are known and there are many scientific reports on supporting therapy with probiotic bacteria with neuromodulatory potential, there is still a lack of systematic knowledge about the species of psychobiotic bacteria, their sources of origin, and the produced metabolites that act neuromodulatory in the human body. This information may be helpful in developing effective preventive therapy and depression prophylaxis using psychobiotic preparations. These preparations could also act as a support in patients with depression treated pharmacologically, which would allow for the use of milder doses of psychotropic drugs.

The research question for this systematic review is as follows: What are the effects of psychobiotic microorganisms on mental health outcomes, and what are the key sources, metabolites, and treatment modalities associated with their therapeutic potential? This question aligns with the objective of systematically reviewing and analyzing the current evidence on the role of psychobiotic microorganisms in depression treatment. Specifically, the aims of this review are to identify the primary sources of psychobiotic microorganisms used in mental health research, including specific probiotic strains, prebiotics, synbiotics, and fermented foods; examine key metabolites produced by psychobiotic microorganisms—such as short-chain fatty acids and neurotransmitter-related compounds—and explore their mechanisms of action within the gut–brain axis; evaluate the efficacy of psychobiotic interventions in alleviating core depressive symptoms—particularly anhedonia, anxiety, and cognitive dysfunctions—in both clinical and preclinical studies; and assess the safety and therapeutic potential of psychobiotic-based treatments for mental health disorders. By addressing these objectives, the review will provide a clear understanding of the therapeutic potential of psychobiotics, highlight gaps in the literature, and inform future research and clinical practices related to mental health treatments.

## 2. Methods

### 2.1. Protocol

The systematic review methodology followed the guidelines outlined in the Preferred Reporting Items for Systematic Reviews and Meta-Analysis Protocols (PRISMA) statement [27]. To elucidate the thematic structure of the literature, co-occurrence analysis of keywords was conducted using the VOSviewer (version 1.6.20) tool [28].

### 2.2. Eligibility Criteria

To qualify for inclusion, studies had to examine psychobiotic microorganisms and their impact on depression and other stress-related mental health disorders. Eligible studies included participants with diagnosed depressive disorders, anxiety disorders, stress-related disorders, sleep disorders, or cognitive impairment, as well as undiagnosed individuals exhibiting symptoms of stress-related psychiatric disorders. Animal studies were included if they directly contributed to understanding psychobiotic mechanisms or their effects on mental health-related behaviors. Studies were required to provide detailed information regarding specific psychobiotic strains used, target mental health conditions, mechanisms of action, and validated measurement tools for mental health outcomes. Studies not published in English and publications available only as abstracts, review articles, and commentaries without original empirical data were excluded from the analysis.

### 2.3. Information Sources and Search Strategy

A systematic literature review was conducted using leading electronic scientific databases: Web of Science, Scopus, and PubMed. It is important to note that psychobiotics is a relatively new area of research, with several high-quality studies and narrative reviews on the topic published over the past decade [29,30,31]. However, to ensure the currency of the analyzed data, a five-year publication timeframe (2020–2024) was adopted. Inclusion criteria encompassed both clinical studies conducted on humans and experimental models using laboratory animals. The methodology applied allowed for the identification of the most current and relevant scientific reports in the studied area, providing a comprehensive review of contemporary knowledge in the field. The literature search methodology used a set of keywords identified by the research team. The search algorithm used was as follows: ((psychobiotic OR probiotic OR synbiotic) AND (metabolites OR neurotransmitters) AND (mental health OR mental disorders)) AND NOT (review). The structure of the algorithm was adapted to the requirements of the individual databases searched. One member of the research team downloaded the Research Information System (RIS) files generated by each of the databases searched, which were then imported into the Rayyan^®^ web application, a specialized tool for conducting systematic reviews [32].

### 2.4. Selection Process

Articles for the review were selected in two stages. Initially, a preliminary selection was made based on the analysis of titles and abstracts, verifying their compliance with established criteria. Subsequently, an in-depth evaluation of the full texts of publications qualified in the first stage was conducted to definitively confirm their compliance with the inclusion criteria. Each publication was independently assessed by a team of three researchers. Any ambiguities or differences in opinions regarding the inclusion of a given article were resolved through consultations and substantive debate among the authors.

### 2.5. Data Extraction

Extracted data included the first author, year of publication, study design, and methodological details, such as randomization method, population characteristics (species, gender, and age), and the dysfunction studied (e.g., depression and anxiety). Information was also collected on the bacterial strains used in this study, source, and form of administration. The extracted outcomes primarily focused on behavioral measures (e.g., forced swimming test), while secondary outcomes included bacterial metabolites/neurotransmitters (SCFAs, specific acids such as isobutyric acid and brain neurotransmitters), mechanisms of action (remodeling of gut microbiota, reduction in inflammation, and inhibition of HPA axis hyperactivity), and health benefits (reduction in anxiety-like behaviors and reduction in inflammatory markers). Three authors independently extracted data from each accepted study using the Rayyan^®^ platform. Discrepancies between authors were automatically identified through the platform’s blind review system (blind on/off function), systematically documented, and resolved by consensus with another independent author following predefined protocols for systematic review methodology.

### 2.6. Risk of Bias in Individual Studies

Three authors were trained in the use of the Revised Cochrane Risk-of-Bias Tool for Randomized Trials (ROB 2) according to official training materials to ensure consistent and reliable assessments. The three authors independently assessed risk for each study using the Cochrane Risk of Bias Tool (ROB 2) developed by Sterne [33]. The ROBVIS tool [34] was used to visually present the risk assessment results. Following the ROB 2 methodology, five parameters were assessed independently: (D1) risk related to the randomization process, (D2) risk related to deviations from planned interventions, (D3) risk related to missing outcome data, (D4) risk related to the way in which outcomes were measured, and (D5) risk related to the selection of reported outcomes. The ROB 2 tool classified the overall risk of bias using colors: red indicated high risk, yellow uncertain risk, and green low risk. Any discrepancies in the assessments were resolved by joint discussion between the three authors until consensus was reached.

### 2.7. Data Synthesis

Due to the considerable heterogeneity of the collected data and the methodological diversity of the included studies, it was not possible to conduct a formal meta-analysis. Instead, a qualitative narrative synthesis approach was used. The main results and characteristics of the studies were presented graphically using pie charts, which show the percentage distribution of key variables and observations. This form of visualization allowed for a clear presentation of the proportions of individual categories in the analyzed data, which facilitates the interpretation of the main trends and patterns observed in the collected research material.

## 3. Results

### 3.1. Summary of Studies

The study selection process, conducted in accordance with the PRISMA 2020 guidelines, identified a total of 369 records across three databases: PubMed (*n* = 177), Web of Science (*n* = 64), and Scopus (*n* = 128) (Figure 1). Following the removal of 99 duplicates, 270 unique publications were deemed eligible for the initial screening stage, of which 154 were excluded based on an analysis of titles and abstracts. Consequently, 116 reports were selected for full-text evaluation; however, two were inaccessible, leaving 114 to be assessed for compliance with the inclusion criteria. At this stage, 69 papers were excluded for the reasons shown in Figure 1. Ultimately, 45 studies that met all the eligibility criteria were included in the systematic review.

VOSviewer software was used to perform a co-occurrence analysis of keywords in order to reveal the thematic structure of the literature. The concept map (Figure 2) delineated four principal thematic clusters, each representing distinct research approaches and areas of interest within the subject under investigation. The first cluster (green) is centered on preclinical research, encompassing concepts such as the brain, neurotransmitter, behavior, GABA, brain-derived neurotrophic factor (BDNF), model, and SCFAs. Terms related to animal models (such as rats) and neurobiological processes (such as expression of neural signaling molecules) were prominently associated within this cluster. The second cluster (blue) comprised terminology characteristic of neurodegenerative disease research, with a pronounced emphasis on terms such as mouse, Alzheimer’s, cognitive impairment, and neuroinflammation. The third cluster (red) pertains to clinical and intervention approaches, featuring dominant terms such as patient, group, trial, symptom, efficacy, and probiotic supplementation. The fourth cluster (yellow) represents a central conceptual node with dominant terms such as major depressive disorder, gut dysbiosis, psychobiotics, and serotonin, functioning as an integrative bridge between neurobiological mechanisms and clinical interventions. Analysis of the layout revealed that the centrality of the yellow cluster highlights the key role of depressive disorders and gut dysbiosis as a common denominator connecting different research approaches, suggesting the need for an interdisciplinary approach in future research on the gut–brain axis.

The characteristics of the articles included in this review are detailed in Table 1. Among the bacterial genera used as psychobiotics in depression research (Figure 3A), *Lactobacillus* was predominantly used (45.5%), followed by *Bifidobacterium* (29%), while other genera such as *Bacillus* (7.5%), *Akkermansia* (7.5%), *Enterococcus* (6%), *Streptococcus* (1.5%), *Christensenella* (1.5%), and *Lactococcus* (1.5%) were employed less frequently. Regarding the sources of bacterial strains (Figure 3B), a significant portion remained unspecified (35%), while commercial sources constituted 24% of the strains. Human-derived and food-derived strains were equally represented (16% each), with laboratory collections accounting for the remaining 9%. This distribution reflects the diverse origin of psychobiotic strains and indicates potential areas for more detailed reporting in future studies. The forms of bacterial preparations in psychobiotic research (Figure 3C) showed that live and heat-treated bacteria were used in equal proportion (37.8% each), followed by freeze-dried preparations (15.6%); a small percentage (8.8%) did not specify the physical state. These results underscore the growing interest in both viable and non-viable bacterial preparations for psychobiotic applications. For administration forms (Figure 3D), bacterial suspensions were most frequently used (31.1%), followed by powder forms (22.2%) and commercial formulations (20.0%). Liquid formulations accounted for 11.1%, food-based delivery systems represented 6.7%, and non-specified forms constituted 8.9% of the interventions. This variety of delivery methods highlights the field’s exploration of optimal administration approaches for psychobiotic efficacy. The analysis of mechanisms of action of psychobiotics in depression (Figure 3E) revealed that neurotransmitter regulation and gut microbiota modulation were the predominant pathways (27.1% each), followed by HPA axis and stress response mechanisms, which accounted for 13.6% of the studied pathways, while SCFAs and metabolite production represented 16.9%. Inflammation and immune regulation mechanisms constituted 15.3% of the reported mechanisms, highlighting the multifaceted nature of psychobiotic action on the gut–brain axis in the context of depressive disorders.

An analysis of the studies regarding the employed methodology and research instruments—specifically, the psychological tests—is provided in Appendix A.

### 3.2. Quality Assessment—Risk of Bias

The risk of bias in the included studies, shown in Figure 4 and Appendix A, indicates that the overall risk assessment was of some concern. Analysis of the domain related to the randomization process (D1) showed that most studies were at low risk of bias, with only a small proportion of studies having some concern. The distribution was similar for the domain related to deviations from the intended intervention (D2). The domain of missing outcome data (D3) showed low risk of bias in the vast majority of studies, but high risk was also found in a few cases. For outcome measures (D4), the vast majority of studies were assessed as having a low risk of bias, with a small proportion having some concern and a similar proportion having a high risk. The selection of reported outcomes (D5) was at low risk of bias in about two-thirds of studies, while the remainder was divided between some concern and high risk, with the former predominating. Assessment of the overall risk of bias revealed that just over half of the studies were at low risk, about a quarter were of some concern, and the remainder were at high risk of bias.

## 4. Discussion

By synthesizing the analyzed evidence, this review aimed to determine the therapeutic potential of psychobiotics in the context of treating depressive disorders.

The systematic review of psychobiotic studies allowed us to identify specific bacterial species that demonstrate the greatest therapeutic potential in the context of mental health and to point to their natural sources of origin, which has significant implications for nutritional strategies. Our analysis reveals that among the microorganisms with the most documented psychobiotic effects, *Lactobacillus plantarum* stands out. Strains of this species (JYLP-326, CR12, P72, 299v, and GM11) have been shown in numerous studies to be effective in alleviating symptoms of depression, anxiety, and cognitive dysfunction [58,68,71,77]. The mechanism of action of *L. plantarum* includes modulation of the gut–brain axis by regulating the level of neurotransmitters, in particular serotonin and GABA, and the production of SCFA. Other research studies have also confirmed that *L. plantarum* is widely present in non-dairy fermented products and has the ability to rapidly reduce symptoms of depression [80].

Our analysis demonstrates that another species with significant psychobiotic potential is *Bifidobacterium breve*, especially the CCFM1025 strain, which in clinical studies has shown the ability to alleviate symptoms of depression and insomnia, mainly by modulating tryptophan metabolism and regulating the serotonergic system [48,54,68]. Based on our systematic evaluation, we observed that the effectiveness of this strain compared to other psychobiotics is particularly high in the case of sleep disorders, which may be a promising direction for further clinical research. Other studies also confirm that *B. breve* occurs naturally in dairy products and breast milk, demonstrating immunomodulatory properties and neurodevelopmental benefits [81,82]. Mosquera et al. also showed in their studies that psychobiotics are particularly effective in reducing the symptoms of depression, with several strains, especially *B. breve* CCFM1025 and combinations of *Lactobacillus* and *Bifidobacteria*, showing significant therapeutic efficacy [83].

The results of our analysis indicate a novel link between *Akkermansia muciniphila* and mental health. Of particular note are its antidepressant and procognitive effects, which are associated with the regulation of serotonin pathways, protection of gut barrier integrity, and anti-inflammatory properties. The patterns identified in our analysis provide new insights into the mechanisms by which this microorganism may affect the gut–brain axis and potentially be part of the treatment of mood disorders [55,57,60].

*Lactobacillus rhamnosus* (strains zz-1, UBLR-58, and JB-1) and *Bifidobacterium longum* complete the list of microorganisms with a documented effect on mood disorders, acting through regulation of the HPA axis and modulation of signaling pathways related to BDNF [28,62,65]. Sarkar et al. additionally point to *Lactobacillus helveticus*, present in fermented milk products such as cheese and yogurt, as a strain with anxiolytic and antidepressant properties via modulation of the gut–brain axis [84].

Live bacterial cultures [35,42] and heat-treated bacteria [66,70] were used equally often, which is an important observation, suggesting that not only live microorganisms but also their inactivated components may have a beneficial effect on mental health. According to the data synthesized in our review, an important observation is the use of both single strains [55,56,57] and complex probiotic formulations containing from several to a dozen or so different bacterial strains [39,52,76], which indicates potential benefits from the synergistic action of different microorganisms. In terms of forms of administration, bacterial suspensions [36,38] and powder forms [44,71] dominated, with the suspensions prepared in various solutions (PBS, water, and culture media), and commercial formulations included capsules and ready-made multi-strain preparations [42,77].

The analyzed studies indicate various natural sources of strains with psychobiotic effects. A significant group consists of traditional fermented milk products, from which effective strains of *S. thermophilus* and *L. plantarum* were isolated [38]. Particularly noteworthy is the traditional Sayram Ketteki yogurt from the Xinjiang region in China, which is the source of *L. plantarum* R6-3 with documented antidepressant effects [42]. Regional fermented plant foods also constitute a rich source of potential psychobiotics. *L. brevis* DL1-11 from the Chinese fermented food pao cai has shown anxiolytic and sleep-enhancing properties [65]. Similarly, *L. plantarum* JYLP-326 from fermented glutinous rice [71] and *L. plantarum* GM11 from Sichuan bean paste [46] have shown antidepressant and anxiolytic effects. The human microbiome is also an important source of strains with psychobiotic potential. *B. breve* CCFM1025 isolated from the feces of a healthy Tibetan adult [48], *L. reuteri* ATG-F4 obtained from fecal samples of Korean newborns [44], and *C. minuta* DSM 32891 from the microbiome of a healthy volunteer [50] are examples of strains with documented effects on neurological functions and behavior.

The studied psychobiotic bacteria affect the gut–brain axis through several main mechanisms. Their ability to modulate the synthesis and metabolism of neurotransmitters plays a key role. *Lactobacillus* and *Bifidobacterium* strains increase the levels of serotonin, GABA, dopamine, and noradrenaline in the brain, which directly affects cognitive functions and emotional state [35,43,59]. The regulation of the HPA axis is particularly important, leading to a decrease in the level of corticosterone/cortisol, as shown in studies with *A. muciniphila* and *L. plantarum* [50].

The production of SCFAs, especially butyrate, acetate, and propionate, is another important neuromodulatory mechanism. These metabolites exhibit neuroprotective and anti-inflammatory effects. Psychobiotics such as *B. breve* CCFM1025 also modulate tryptophan metabolism by influencing the kynurenine and indole pathways, which translates into the regulation of serotonin levels [47,48,54].

Based on our systematic evaluation, we observed that an important aspect of the action of psychobiotic bacteria is their influence on the expression of the neurotrophic factor BDNF, which is crucial for neuroplasticity and cognitive function [36,42,79]. Additionally, they exhibit anti-inflammatory effects by reducing the level of pro-inflammatory cytokines (TNF-α and IL-1β) and increasing anti-inflammatory cytokines (IL-10) [50,58]. Protection of the integrity of the intestinal barrier by regulating tight junction proteins prevents the penetration of endotoxins into the bloodstream, which also contributes to the reduction in neuroinflammation [46,59].

The assessment of the effectiveness of psychobiotics in the treatment of depressive disorders based on the analyzed studies indicates promising results. Clinical studies involving patients diagnosed with depressive disorders showed statistically significant improvement in depression scales (HDRS, MADRS, and BDI-II) after the use of psychobiotics, especially *B. breve* CCFM1025 and *L. plantarum* 299v [70,76,77]. Similar results were obtained by Kazemi et al., where the combination of *L. helveticus* and *B. longum* led to a significant reduction in depressive symptoms measured by the BDI compared to placebo [85]. Particularly pronounced effects were observed in animal models of depression, where administration of psychobiotics resulted in a reduction in depressive and anxious behaviors [42,43,56].

An interesting observation is the potential of psychobiotics as a complementary therapy. A study with a combination of *L. helveticus* R0052 and *B. longum* R0175 showed an improvement in BDI-II scores in patients taking antidepressants [79]. It should be noted, however, that not all studies show clear results. Reininghaus et al. (2020) and Romijn et al. (2017) did not observe statistically significant differences between the groups receiving psychobiotics and placebo [86,87]. However, it is worth noting the varied clinical response, suggesting that the effectiveness may depend on the individual composition of the patient’s microbiome [76]. Additionally, psychobiotics have shown a beneficial effect on symptoms accompanying depression, such as sleep disorders [72,74] and gastrointestinal complaints [70,73].

Our systematic evaluation indicates that traditional fermented foods may serve as natural sources of psychobiotics, supporting their potential role in the daily diet. The synthesis of available evidence further supports the promotion of regional products such as yogurt, kefir, kimchi, and various pickled vegetables as a simple strategy to enrich the gut microbiota with beneficial strains that support the gut–brain axis.

Our review identifies that although the research results are promising, it should be noted that clinical trials on psychobiotics for the treatment of depressive disorders are still in the early stages, characterized by small study groups and diverse methodology. However, current evidence suggests significant potential for psychobiotics both as an adjunct to conventional antidepressant therapies and as a component of the prevention of mood disorders through an appropriately composed diet. Based on our integrative approach, we identified that future research should focus on larger, methodologically sound randomized controlled trials with well-defined outcome measures and longer follow-up periods. Our analysis clearly highlights that, in light of the presented data, the development of functional food products enriched with strains with documented psychobiotic effects may be a promising strategy for supporting mental health through nutritional interventions.

### 4.1. Limitations of the Studies Included in the Review

Despite the encouraging findings, certain methodological limitations should be acknowledged. The diversity in psychological tests employed in these studies significantly affects the comparability of research outcomes. In animal model research, while the Open Field Test (OFT), Forced Swimming Test (FST), and Elevated Plus Maze (EPM) were frequently used, the incorporation of a broad array of additional tests complicates direct comparisons because these tests measure different aspects of behavior, such as locomotion, anxiety, or depressive-like states [36,39,41]. This variability makes it challenging to integrate findings, as the operational definitions of outcomes may differ between tests.

Similarly, in human studies, the usage of multiple scales—ranging from the Hamilton Depression Rating Scale (HDRS) to various sleep quality, anxiety, and additional symptom scales—reflects a lack of standardization in assessing depressive disorders and related symptoms [70,71,73]. The small sample sizes for each specific scale further limit the statistical power and robustness of cross-study comparisons.

Overall, the heterogeneity of assessment tools in both preclinical and clinical research underlines the need for standardized methodologies. Establishing a common set of validated tests or scales would enhance consistency, facilitate meta-analyses, and ultimately improve the translational value of psychobiotic research in depressive disorders. To enhance standardization and comparability of intervention outcomes, greater consensus on behavioral measures for cognitive and mental health is needed. This aligns with recent initiatives by major research funders, such as the NIMH (Bethesda, MD, USA) and Wellcome Trust (London, UK), advocating for more unified approaches in mental health research [88].

The overall risk of bias in the included studies raises some concerns regarding the reliability of their findings. While the majority of studies demonstrated a low risk of bias in domains such as the randomization process (D1) and deviations from intended interventions (D2), there were notable exceptions in other areas. Specifically, three studies showed issues with missing outcome data (D3), four studies exhibited high risk in the measurement of outcomes (D4), and two studies had high risk in the selection of reported results (D5). These findings suggest that although many studies are methodologically sound in certain areas, attention should be paid to the domains with higher risks when interpreting their outcomes.

### 4.2. Implications of the Results for Practice and Policy

The findings of this review suggest potential applications in food science, nutrition, and public health, particularly in the development of functional foods, dietary supplements, and foods for special medical purposes (FSMPs) aimed at supporting mental well-being. The frequent use of *Lactobacillus* and *Bifidobacterium* strains—many of which are naturally present in fermented foods—indicates a feasible pathway for incorporating psychobiotics into everyday diets and commercial products.

The observed use of both live and heat-treated strains highlights technological flexibility, offering options for improved stability and product design. Additionally, the identified mechanisms of action, such as neurotransmitter modulation and gut microbiota regulation, support the concept of psychobiotics as modulators of the gut–brain axis.

For industry and policymakers, these results emphasize the importance of validated strains, accurate labeling, and evidence-based health claims. Future efforts should focus on translating this evidence into safe, effective, and accessible interventions while addressing regulatory and consumer expectations.

## 5. Conclusions

This systematic review synthesizes current evidence on the use of psychobiotics in the context of depression, with a focus on bacterial sources, metabolites, mechanisms of action, and clinical outcomes. Among the 45 included studies, strains from the genera *Lactobacillus* and *Bifidobacterium* were most frequently investigated, accounting for nearly 75% of the total, with some strains—such as *L. rhamnosus* JB-1 and *B. longum* 1714—demonstrating notable psychotropic effects. The majority of interventions utilized strains derived from food or human sources, and both live and heat-treated preparations were found to be comparably represented, suggesting growing interest in diverse formulation strategies.

The main mechanisms underlying psychobiotic efficacy included modulation of neurotransmitters (e.g., GABA and serotonin), regulation of the HPA axis, production of SCFA, immune modulation, and restoration of gut microbiota balance. These pathways highlight the complex and multifactorial nature of psychobiotic action on the gut–brain axis.

While clinical outcomes varied, several studies reported significant improvements in depressive symptoms, anxiety, and stress-related markers in both animal models and human subjects. However, the overall heterogeneity in study design, strain specificity, dosage, duration, and psychological assessment tools presents a challenge for drawing definitive conclusions.

In summary, psychobiotics hold promising potential as adjunctive or preventive tools in the management of depressive disorders. Their integration into dietary supplements, functional foods, and medical nutrition strategies is a realistic goal, provided future research continues to address current methodological limitations. Standardization of clinical protocols, long-term safety assessments, and real-world trials in food-based applications will be crucial to advancing the therapeutic potential of psychobiotics and translating microbiome science into meaningful mental health solutions.

## Figures and Tables

**Figure 1 nutrients-17-02139-f001:**
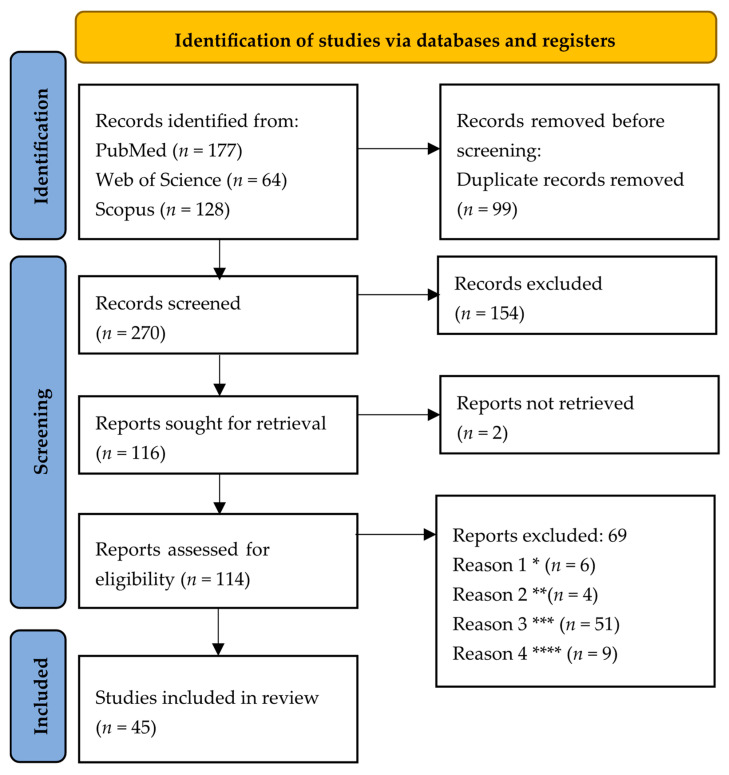
PRISMA flow diagram summarizing the process of article screening and reasons for exclusion. * Reason 1—Research limited to microbiome analysis, ** Reason 2—Dietary studies with an intervention profile that does not meet eligibility criteria, *** Reason 3—Studies on individuals with disorders other than MDD, **** Reason 4—Studies using only subjective measurement tools or research protocols in the planning phase.

**Figure 2 nutrients-17-02139-f002:**
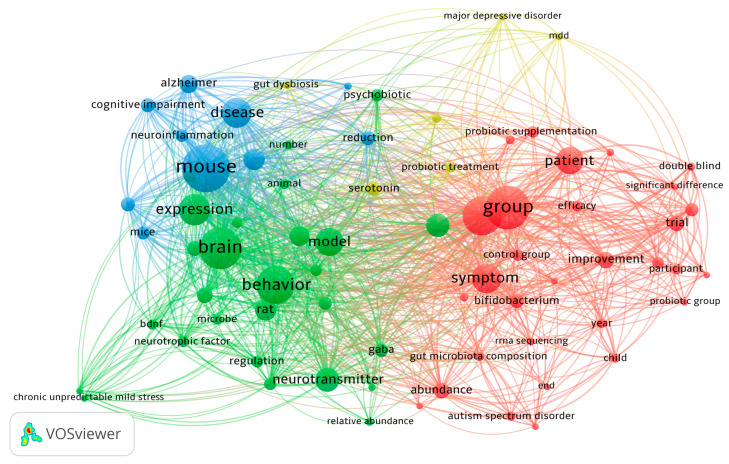
Analysis of the interaction between psychobiotic strains and depression: keyword and abstract co-occurrence network using VOSviewer.

**Figure 3 nutrients-17-02139-f003:**
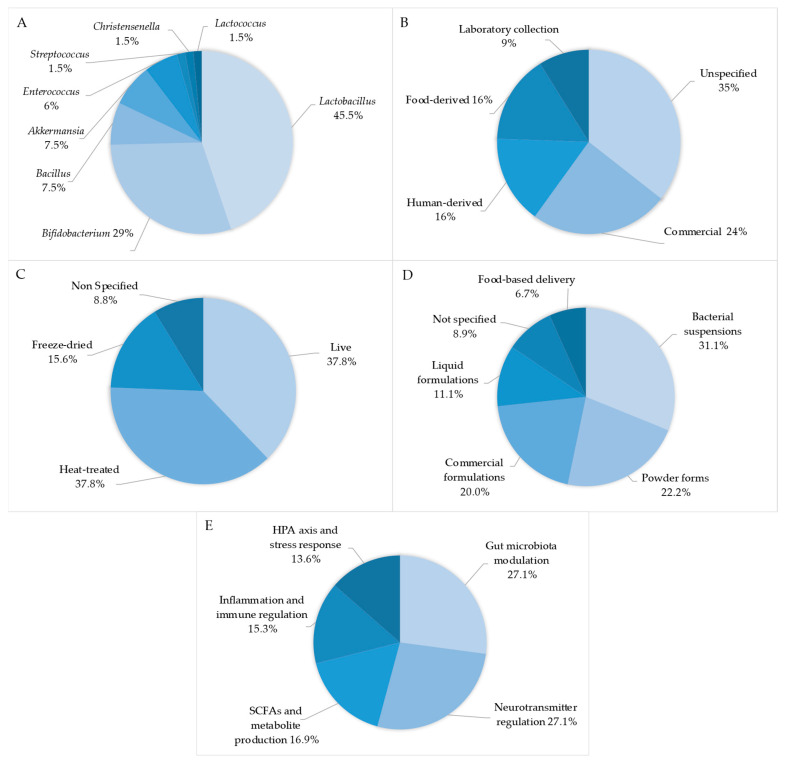
Characteristics of psychobiotics used in research on depressive disorders: (**A**) percentage distribution of bacterial genera, (**B**) sources of bacterial strains, (**C**) forms of bacterial preparations, (**D**) administration forms of psychobiotics, and (**E**) percentage distribution of psychobiotic mechanisms of action in depressive disorders.

**Figure 4 nutrients-17-02139-f004:**
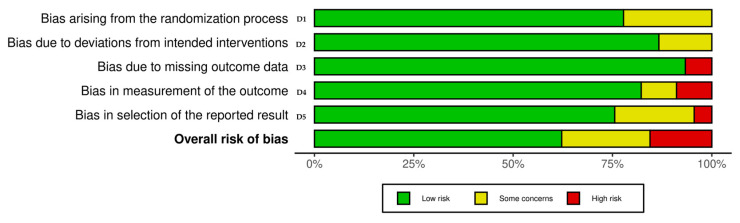
Risk of bias assessment for included studies.

**Table 1 nutrients-17-02139-t001:** Characteristics of studies included in the systematic review.

Study ID		Population/Sample Size	Dysfunction	Intervention-Bacteria				Outcomes		
Animal Studies										
Author	Study Design			Species/Strain	Source of Origin	Form/Dose/Time of Treatment	Behavioral Tests	Bacterial Metabolites/Neurotransmitters	Mechanism	Health Benefits
Feng et al., 2022 [35]	RCEAS	Male Sprague Dawley rats, *n* = 68	Subhealth state induced by chronic stress and antibiotics	*B. licheniformis* BL20386	Vaginal swab	Live bacte- rial suspen- sion/1 mL, 1 × 10^8^ CFU/mL)/ 5–8 weeks	FST, EPM	SCFAs, isobutyric acid, isovaleric acid, propionic acid, GABA, 5-HT, Glu, NE	Gut microbiota modulation; anti-inflammation; HPA axis inhibition	↓anxiety behaviors, ↓inflammatory markers, ↓CORT ↑gut microbiota,
Xu J. et al., 2022 [36]	RCEAS	Male C57BL/6 mice, *n* = 60	Depression induced by CUMS	*L. rhamnosus* zz-1	Northeast Agricultural University, Harbin, China	Bacterial suspension/0.1 mL, 2 × 10^7^–10^9^ CFU/kg bw)/ 5 weeks	SPT, FST, OFT Body weight monitoring	5-HT, NE, DA	HPA axis inhibition; BDNF/TrkB activation; microbiota modulation	↓depression, ↓inflammation, ↑intestinal barrier, ↑gut microbiota, ↑neurotransmitters
Zhao et al., 2022 [37]	EIS	Rhesus monkeys, *n* = 30	SD-induced stress responses and gut dysbiosis	Bifid Triple Viable Capsules: *B. longum*, *L. acidophilus*, and *E. faecalis*	-	Capsules—420 mg in water/2×/day/30 days	Paradoxical sleep disruption, light and noise response	GABA, NE	Gut microbiota modulation; GABA elevation; gut–brain axis regulation	↓stress hormones, ↓inflammation, ↑gut microbiota
Kim et al., 2022 [38]	EIS	Male SPF C57BL/C mice, *n* = 11–12/group, 3 groups	Cognitive function and memory	*S. thermophilus* EG007, *L. plantarum* A003F7	Fermented dairy products	Suspension in water/0.2 mL (1.08 or 3.12 × 10^9^ CFU)/4 weeks	Y-SAT, Y-FAT, NORT, PAT	GABA	GABA production; gut microbiota modulation	↑spatial memory, ↑learning, ↑object recognition
Dandekar et al., 2022 [39]	EPAS	Sprague-Dawley rats, *n* = 6–8/group, 3 groups	Depression and anxiety-like behaviors induced by MS and CUMS	Probiotic: *B. coagulans* Unique IS-2, *L. plantarum* UBLP-40, *L. rhamnosus* UBLR-58 *B. lactis* UBBLa-70, *B. breve* UBBr-01, *B. infantis* UBBI-01	Commercial multi-strain probiotic (Cognisol, Unique Biotech Ltd., Telangana, India)	Freeze-dried probiotic in water/1 capsule daily/6 weeks	FST, SPT, EMP, OFT	5-HT, DA, Acetate, Propionate, Butyrate	Gut–brain axis modulation; Trp metabolism regulation	↓anxiety/depression, ↓neuroinflammation, ↑neurotransmitters, ↑intestinal function
Alizadeh et al., 2022 [40]	ECAS	Male Wistar rats, multiple groups of 7 rats	Stress-induced memory loss/amnesia	*L. plantarum* ATCC 8014, *L. brevis* ATCC 14869, *B. bifidum* ATCC 29521	-	Daily suspen- sion in water/2 × 10^6^–10^9^ CFU/2–5 weeks	PALT, STL, EPST	GABA	GABAergic system modulation; BDNF expression regulation	↓memory loss, ↑memory formation, ↑signaling pathways
Carlessi et al., 2022 [41]	EAS	Male and female Wistar rats, *n* = 12–15/group, 5 groups	MDD induced by maternal deprivation	*B. infantis*	-	Probiotic/1 × 10^10^ CFU in 100 mL water/10–50 days	OFT, FST, ST	-	Oxidative damage regulation; blood–brain barrier (BBB) protection	↓depression, ↓oxidative stress markers, ↑BBB integrity
Zhao et al., 2023 [42]	EAS	Male C57BL/6J mice, 4 groups	Depression (CUMS-induced depression)	*L. plantarum* R6-3	Sayram Ketteki (Xinjiang fermented yogurt)	Live bacteria/5 × 10^9^ CFU in 0.9% saline/8 weeks	SPT, TST, FST Body weight measurements	SCFAs, 5-HT, DA, NE	Gut–brain axis signaling; SCFA production; BDNF/TrkB pathway activation	↓depression, ↑weight, ↑hippocampus, ↓CORT levels, ↑BDNF
Hu et al., 2023 [43]	EAS	Male SD rats, *n* = 7/group, 6 groups	Depression (CUMS-induced depression-like behavior)	*L. plantarum* (M618)	Laboratory-preserved strain	Fermented wheat germ powder/ 43–60 mg GABA/kg/day/4 weeks	EPM, FST, OFT, SPT	5-HT, 5-HIAA, ACH, GABA	Gut microbiota structure restoration; amino acid metabolism regulation	↓depression, ↑neurotransmitters, ↑weight, ↓anhedonia
Lee et al., 2023 [44]	ECAS	C57BL6/N male mice, *n* = 50	Chronic stress-induced anhedonia	*L. reuteri* ATG-F4 (F4)	Fecal samples of newborn babies in Daejeon, South Korea	Live bacteria/ 5 × 10^8^ CFU/mL/ 250 µL daily/ 4 weeks	SPT, body weight measurements	5-HT, Trp metabolites, 5-HTP, 5-HIAA	Trp metabolism modulation; serotonin regulation; gut microbiota modulation	↓anhedonia, ↑weight, ↑serotonergic system
Feng et al., 2023 [45]	ECAS	Male Sprague Dawley rats, *n* = 52	Depression and anxiety induced by CUMS	*B. licheniformis* CGMCC NO.0298	Zhengchangsheng^®^ (Liyang, China)	Live bacteria/ 1 × 10^8^ CFU/mL/1 mL daily/ 4 weeks	FST, EMP	GABA, Trp, DA, epinephrine, SCFAs	Gut microbiota modulation; SCFA production; neurotransmitter regulation	↓depression, ↓anxiety, ↑gut microbiota
Ma et al., 2023 [46]	EAS	Adult female C57BL/6J mice, *n* = 10/group	Depression induced by CUMS	*L. plantarum* CR12	-	Live bacteria in powder form/ 1 × 10^9^ CFU/mL/ 200 μL daily/ 3 weeks	OFT, FST, TST, SPT, MWM	SCFAs, butyrate, 5-HT, DA, NE	Gut microbiota modulation; butyrate enhancement; neuroinflammation reduction	↑cognitive function, ↓anxiety/depression, ↑gut barrier
Galley et al., 2024 [47]	EIS	Female nulliparous C57Bl6/J mice and offspring, *n* = 82	Prenatal stress (PNS) exposure	*B. dentium* 27678	American Type Culture Collection (ATCC)	Live bacteria in 0.9% saline/3.0–9.0 × 10^8^ CFU/150 μL daily/7 days	TCSBT, LDT, SB	KA, I3PA, 5-HT, Trp metabolites	Anti-inflammatory action; Trp metabolism modulation	↓maternal inflammation, ↓fetal neuroinflammation, ↑social behavior
Qian et al., 2024 [48]	EAS	Male C57BL/6J mice, *n* = 69	CUMS-induced anxiety, depression-like behavior, and memory impairment	*B. breve* CCFM1025	feces of a healthy adult Tibetan man	Live bacteria in suspension/ 1 × 10^7^–10^9^ CFU/ oral daily/ 7–35 days	OFT, EPM, FST, LDB	CA, IAM, IAA, ILA, Trp metabolites 5-HT	Gut metabolite modulation; bile salt hydrolase activation; HPA axis regulation	↓anxiety, ↑memory, ↓depression, ↑gut homeostasis
Lozano et al., 2024 [49]	ECIS	Adult male Wistar rats, *n* = 12/group, 2 groups	Anxiety- and depression-like behaviors	*L. plantarum* LPB145	Uruguayan cheese starter isolate	Lyophilized live bacteria in skim milk/ 5 × 10^8^ CFU/ 0.5 mL daily/ 28 days	EMP, FST, OFT	GABA	GABA production; specific microbiota enhancement	↓depression behaviors, no effect on anxiety/locomotion
Agusti et al., 2024 [50]	EIS	Male C57BL/6 mice and CD-1 mice as aggressors, *n* = 45/group, 3 groups	SD-induced disturbances	*C. minuta* DSM 32891	feces of a healthy volunteer	Live bacteria suspended in PBS with 0.05% L-cysteine + 10% glycerol/1 × 10^9^ CFU daily/6 weeks	FST, TST, SPT, OFT, LDT, SIT	DA, NA, ADE, CORT, DOPAC, HVA, 3-MT	HPA axis modulation; dopamine metabolism regulation; receptor expression modulation	↓depression/anxiety, ↓inflammation, ↓cardiovascular damage
Guo et al., 2023 [51]	EIS	Male C57BL/6J SPF mice, NIAAA: *n* = 12/4 groups, CUMS: *n* = 15/3 groups	Depression-like behaviors induced by: Chronic alcohol exposure, CUMS	*A. muciniphila* (ATTC BAA-835)	-	Live bacteria in 5% glycerol/ 2.5 × 10^9^ CFU/ 200 μL daily/ 4 weeks	FST, TST, SPT, OFT, FC	5-HT	Serotonin enhancement; SERT expression inhibition	↓depression, ↑hedonic response, ↑liver function, ↑serotonin
Zhang et al., 2024 [52]	EIS	Male Sprague-Dawley rats, *n* = 30 rats	SD-induced anxiety-like behaviors	*B. longum* *L. acidophilus* *E. faecalis* *B. licheniformis* *B. subtilis* *E. faecium*	Commercial probiotics	Live multi-strain bacteria in saline/ 1 × 10^6^–10^8^ CFU per strain/daily for 14 days	OFT	LPS, serum metabolites, including uridine and Trp	LPS reduction; inflammatory bacteria reduction	↑anxiety behaviors, ↓serum LPS
Xie et al., 2024 [53]	EAS	Male C57BL/6 mice, *n* = 30 mice	EHS-induced cognitive impairment	*L. murinus*, Probiotic blend: *B. bifidum* F-35 *B. longum* CCFM729, *Lactobacillus* species	-	Live multi-strain blend/0.02 mg/kg/daily for 28 days	OFT, EPM, TST, FST, NORT	SCFAs	Microbiota equilibrium restoration; BDNF/TrkB pathway modulation	↑cognitive performance, ↑BDNF/TrkB, ↑cognitive recovery
Tian et al., 2024 [54]	EIS	C57BL/6J mice, *n* = 8/group, 3 groups	SDStress-induced cognitive impairment and circadian rhythm disturbance	*B. breve* CCFM1025	-	Live bacteria in 10% skim milk/1 × 10^10^ CFU/mL/daily for 7 days	OFT, NORT, Y-maze test	Isovaleric acid GABA, serum purine metabolites, 5-HT, melatonin	Gut microbiota modulation; striatal melatonin system regulation	↑cognitive performance, ↑weight/food intake, ↑serotonin metabolism
Du et al., 2024 [55]	EAS	Male db/db mice and db/m controls, *n* = 12/group, 4 groups	Diabetic cognitive impairment (DCI)	*A. muciniphila*	-	Live bacteria in PBS/ 5 × 10^9^ CFU/mL/ oral daily/ 8 weeks	MWM, EL, PCT, TSTQ, SS	GA, Gly, Xyl, IAA	Neuroinflammation reduction; gut microbiota modulation; synaptic structure improvement	↑cognitive function, ↑memory, ↑synapses, ↓neuroinflammation
Chen et al., 2024 [56]	RCEAS	Male BALB/c mice, *n* = 10/group, 4 groups	Depression-like behaviors induced by CRS	*L. lactis* WHH2078	Hangzhou Wahaha Group Co. (Hangzhou, China)	Live bacteria in saline/ 5 × 10^8^ CFU/mL/ 0.2 mL daily/ 3 weeks	TST, OFT, FST, SPT	5-HT CORT	Gut microbiome modulation; microbial diversity restoration	↓depression, ↑exploration, ↑locomotion, ↑gut dysbiosis
Kang et al. 2024 [57]	HBCCS-AE	Cirrhosis pts ± HE; Cognitive dysfunction (*n* = 154); C57BL/6J mice	Hepatic encephalopathy Cognitive dysfunction Liver injury/cirrhosis	*A. muciniphila*	-	Live bacteria/ ~1 × 10^9^ CFU/mL/ 200 μL/ 3×/week/	NORT, WMT, TST, SNSB	5-HT, BDNF	BDNF/5-HT pathway regulation; 5-HT receptor suppression	↑cognitive function, ↓liver injury, ↓gut inflammation
Lee et al., 2024 [58]	EAS	Male C57BL/6 mice, multiple groups of 6–8 mice	Depression/anxiety (DA) Insomnia, Stress-induced inflammation	*L. plantarum* P72	healthy human feces bacterial collection	Live bacteria (P72) in 0.1% trehalose/ 1 × 10^9^ CFU/day/ oral daily/ 5–7 days	OFR, EPM, TST, SLT, SD	GABA, 5-HT, GABAA receptor α1, 5-HT1A receptor	GABA/GABAA receptor upregulation; serotonin/5-HT1A receptor enhancement	↓depression, ↓anxiety, ↑sleep parameters, ↓inflammation
Wei et al., 2024 [59]	ECAS	C57BL/6J male mice, *n* = 10/group,4 groups	Cognitive impairment induced by high-fat diet	*L. kisonensis* (JCM15041)	State Key Lab of Bioreactor Eng. (East China Univ. of Sci.& Tech. Shanghai)	Live bacteria in fermented grain mixture/ 5 × 10^8^ CFU/mL/ 10 mL/kg daily/ 10 weeks	NBT, OFT, EPM	DA, EPI, NE, 5-HT, ACH	Gut microbiota dysbiosis regulation; neurotransmitter pathway upregulation	↑behavioral skills, ↑neurotransmitters, ↓synaptic damage
Ding et al., 2021 [60]	EAS	Male C57BL/6 mice, *n* = 6/group, 3 groups	Chronic restraint stress (CRS)-induced depression-like behavior	*A. muciniphila* (ATCC^®^ BAA-835^TM^)	-	Live bacteria (5 × 10^8^ CFU/mL, 200 µL), oral, daily/3 weeks	OFT, TST, FST	CORT, DA, 5-HT, BDNF	Hormone/neurotransmitter/BDNF regulation; gut microbiota modulation	↓depression, ↑locomotor activity, ↑neurotransmitter levels
Natale et al., 2021 [61]	RCEAS	Male Long-Evans rats, *n* = 6/group, 4 groups	Chronic unpredictable stress (CUS)	*L. helveticus* R0052 (5%) *L. rhamnosus* R0011 (95%)	Lacidofil^®^ probiotic (Mirabel, QC, Canada)	10^9^ CFU/mL, in water, oral, daily/27 days	OFT, FST	CORT, DHEA	Microglia reactivity reduction; DHEA/CORT ratio improvement	↑emotional resilience, ↓anxiety markers, ↑exploration
Westfall et al., 2021 [62]	EIS	Male C57BL/6J mice, *n* = 12–16/group, 8 groups	Chronic stress-induced anxiety and depression-like behaviors	*L. plantarum* ATCC 793 *B. longum* ATCC 15707	-	Live bacteria in drinking water/1 × 10^9^ CFU per strain daily/oral/6 weeks	OFT, FST	4-HPPA, 4-HPAA, CA, 5-HT, kynurenine metabolites	AHR receptor activation; inflammatory response reduction	↓anxiety/depression, ↓neuroinflammation, ↑immune regulation
Wang et al., 2021 [63]	EAS	C57BL/6J male mice, *n* = 36/group, 3 groups	Restraint stress-induced memory dysfunction	*L. johnsonii* BS15 (CCTCC M2013663)	Hongyuan Prairie yogurt (Aba, Taicang, China)	Live suspension (2 × 10^8^ CFU in 0.2 mL PBS), oral daily, 28 days	NORT, T-maze test, Passive avoidance test	DA, 5-HT, Ach, Glu, GABA, nitric oxide (NO)	Memory-related protein enhancement; neurotransmitter level increase	↑memory, ↓CORT, ↑gut barrier, ↑inflammation markers
Liu et al., 2020 [64]	EIS	Male C57BL/6 mice and CD-1 retired breeder mice, *n* = 74 mice	PTSD-like behavior induced by chronic social defeat (CSD)	*L. rhamnosus* JB-1	Alimentary Health Ltd., Cork, Ireland	Live bacteria (1 × 10^9^ CFU in 200 μL PBS), oral daily	3-chamber test, Aggressor avoidance test, OFT, LDT, EMP	CRHR-1 BDNF	CRHR-1 expression reduction; BDNF expression reduction	Negative effects: ↑avoidance behavior, ↑social deficits
Yu et al., 2020 [65]	EAS	Male ICR mice, *n* = 10/group, 6 groups	Insomnia and anxiety	*L. brevis* DL1-11	Traditional Chinese fermented food pao cai	Live bacteria (1 × 10^8^ CFU/mL), oral gavage/30 days	OFT, EPM, SLD	GABA, SCFAs, Butyric acid	Beneficial bacteria abundance increase; SCFA production enhancement	↓anxiety behavior, ↑sleep time, ↓sleep latency
Kambe et al., 2020 [66]	EIS	Male C57BL/6J mice, *n* = 8/group, 2 groups	Anxiety-like behavior	*E. faecalis* EC-12	-	0.125% heat-killed bacteria/4 weeks	OFT, EMP, FST	EC-12	Neurotransmitter receptor gene upregulation; gut microbiota composition alteration	↓anxiety behavior, anti-depressive trend, ↑exploration
Duranti et al., 2020 [67]	EAS	*n* = 32 Male wild-type Groningen rats	Anxiety and depression	*B. adolescentis* PRL2019 *B. adolescentis* HD17T2H	Human gut/intestine/feces	Live bacteria in solution, 1 × 10^9^ CFU/day/5 days	-	GABA	GABA production via GAD enzyme system	↑GABA production, potential gut–brain axis modulation
Ma et al., 2023 [68]	EAS	Male SPF SD rats, *n* = 10/group, 3 groups	Depression induced by CUMS	*L. plantarum* GM11	Sichuanbroad bean paste (fermented food)	Live bacteria (2 × 10^9^ CFU/mL), oral gavage/ 21 days	FST, SPT, EMP, OFT	5-HT, CORT, BDNF	Serotonin and BDNF level increase; CORT reduction	↓depression, ↓despair, ↓anhedonia
Rayan et al., 2024 [69]	EIS	Male Long-Evans rats, 6 groups	MDD and anxiety disorders	Lacidofil^®^ probiotic: *L. rhamnosus* Rosell^®^-11 + *L. helveticus* Rosell^®^-52)	Lallemand, Mirabel, QC, Canada	Probiotic formulation (1.5 × 10^9^ CFU/day)/ 21 days	EPM	-	Synaptic/signaling gene upregulation; prefrontal cortex regulation	↓anxiety, GWAS loci enrichment, ↑neurite branching
**Human studies**										
Tian et al., 2022 [70]	RPDBCT	Adults, *n* = 45	MDD with associated gastrointestinal symptoms	*B. breve* CCFM1025	-	Freeze-dried powder in sachet/ 10^10^ CFU/ 1×/day/ 4 weeks	HDRS, MADRS, BPRS, GSRS	5-HT, 5-HIAA, Trp metabolites, indole derivatives	Gut microbiome regulation; Trp metabolism modulation	↓depression symptoms, ↑ gastrointestinal symptoms, ↑emotional regulation
Zhu et al., 2023 [71]	RCT	College students, *n* = 30/group, 3 groups	Test anxiety, depression, and insomnia	*L. plantarum* JYLP-326	Fermented sticky rice (Bama, Guangxi, China)	Sachet with 1.5 × 10^10^ CFU live bacteria/ oral/ 3 weeks	HAMA, HDRS, AIS	Ethyl sulfate, 1,2-propanediol	Gut microbiota modulation; fecal metabolite regulation	↓anxiety, ↓depression, ↑insomnia, ↑microbiota
Lan et al., 2023 [72]	RCT	Adults, *n* = 40/group, 2 groups	Stress-induced insomnia	*B. breve* CCFM1025	-	Sachet with 1 × 10^9^ CFU live bacteria/ 4 weeks	PSQI, AIS	-	Stress marker reduction by daidzein; serum metabolite modulation	↓sleep quality scores, ↑subjective sleep, ↓sleep disturbance
Wang et al., 2024 [73]	RPDBCT	College students, 100 students	DCS with associated anxiety and depression	*B. breve* BB05	China General Microbiological Culture Collection Center	Live bacteria in sachet/ 1 × 10^10^ CFU per dose/ 2×/day/ 2 weeks	BSS, HAMA, HDRS	5-HT, ACH, EPI, NE	Gut microbial diversity enhancement; beneficial bacteria increase	↓diarrhea symptoms, ↓anxiety/depression, ↑gut microbiota
Li et al., 2024 [74]	RPDBCT	Adults, *n* = 40/group, 3 groups	High mental stress, overweight, insomnia	*B. breve* 207-1	Healthy Chinese infants	Live bacteria in probiotic powder drink/ 1–5 × 10^10^ CFU daily/28 days	SDS, SAS, PSQI	GABA, 5-HT, SCFAs: acetic, propionic, and butyric acids	Gut–brain axis modulation; GABA enhancement; HPA axis hormone suppression	↑sleep quality, ↑diet, ↑exercise, ↓weight, ↑serotonin
Casertano et al., 2024 [75]	RDBPCCS	Adults, *n* = 43–44/group, 2 groups	Mild-moderate stress	*L. brevis* P30021 *L. plantarum* P30025	-	Live bacteria powder stick/ 2 × 10^9^ CFU (+ B6, D3, Zn)/daily/4 weeks	CBB	GABA Acetylcholine Choline Glutamate	Probiotic genera abundance increase	↓depressive symptoms, ↑rumination, no effect on cognitive performance
Kreuzer et al., 2024 [76]	RDBPCT	Adults with MDD, *n* = 24 -intervention group, *n* = 29 (placebo group)	MDD	*B. bifidum* W23, *B. lactis* W51, W52, *L. acidophilus* W22, *L. casei* W56, *L. paracasei* W20, *L. plantarum* W62, *L. salivarius* W24, *L. lactis* W19	OMNi-BiOTiC Stress Repair provided by AllergoSan	Live multistrain bacteria in drink/ 7.5 × 10^9^ CFU oral daily/28 days	BDI-II, HDRS	Butyrate, various amino acids	Clock gene expression modulation; metabolite-clock gene correlation	↑depression scores, ↑circadian rhythm influence
Godzien et al., 2024 [77]	RPDBCT	MDD patients on SSRIs; *n* = 30/group, 2 groups	MDD	*L. plantarum* 299v	Strain owner—Probi AB, Lund, Sweden	Live bacteria in capsule/ 1 × 10^10^ CFU oral daily/8 weeks	-	LCACs, NATs, OxPC, SMs, L-His, D-Val, *p*-cresol	LCACs reduction; sphingomyelin level enhancement	↑cognitive functions, ↑mitochondrial function, ↓inflammatory markers
Quero et al., 2021 [78]	TBRPPS	*n* = 13 soccer players, *n* = 14 sedentary students	Sleep quality, depression, anxiety, stress levels	*B. lactis* CBP-001010 *L. rhamnosus* CNCM I-4036 *B. longum* ES1	-	Live bacteria in synbiotic stick/ 1 × 10^9^ CFU oral daily/4 weeks	HLPCQ, STAI, PSS, BFI, BDI	IL-1β, IL-10, immunoglobulin A, EPI, NE, DA, 5-HT, CRH, ACTH, CORT	Immunophysiological bioregulation; dopamine increase	↑sleep efficiency, ↑perceived health, ↓stress/anxiety, ↓depression
Heidarzadeh-Rad et al., 2020 [79]	RDBPCT	Adults, *n* = 78	MDD	*L. helveticus* R0052 *B. longum* R0175	Lallemand Health Solutions (Mirabel, QC, Canada)	Freeze-dried live bacteria (1 × 10^10^ CFU/day), powder sachet/8 weeks	BDI-II	BDNF	Serum BDNF level increase	↓depression symptoms, ↑BDNF levels, ↓BDI scores

↑—increase/improvement, ↓—decrease/reduction, HE—hepatic encephalopathy, MDD—major depressive disorder, CUMS—Chronic Unpredictable Mild Stress, MS—Maternal Separation, SD—sleep deprivation, EHS—exertional heat stroke, PTSD—post-traumatic stress disorder, FST—Forced Swimming Test, EMP—Elevated plus Maze test, OFT—Open Field Test, SPT—Sucrose Preference Test, Y-SAT Y-maze spontaneous alternation test, Y-FAT Y-maze forced alternation test, NORT—Novel object recognition test, PAT—Passive avoidance test, HDRS—Hamilton Depression Rating Scale, MADRS—Montgomery–Asberg Depression Rating Scale, BPRS—Brief Psychiatric Rating Scale, GSRS—Gastrointestinal Symptom Rating Scale, PALT—Passive avoidance learning task, STL—step-through latency measurements, EPST—Elevated platform stress test, TST—Tail suspension test, ST—Splash test, HAMA—Hamilton Anxiety Scale, AIS—Athens Insomnia Scale, MWM—Morris water maze, PSQI—Pittsburgh Sleep Quality Index, TCSBT—three-chamber social behavior test, LDT—light–dark test for anxiety-like behavior, SB—social behavior, BSS—Bristol Stool Scale, LDB—light dark box, BDI—Beck’s Depression Inventory scores, HLPCQ—The Healthy Lifestyle and Personal Control Questionnaire, STAI—State-Trait Anxiety Inventory, PSS—Perceived Stress Scale, BFI—Brief Fatigue Inventory, NBT—nesting behavior test, SLT—sleep latency time, SLD—sleep duration, EL—Escape latency, PCT—platform crossing times, TSTQ—time spent in target quadrant, SS—swimming speed, FC—conditional fear test, SIT—Social interaction test, CBB—CogState Brief Battery tests, SDS—Self-Rating Depression Scale, SAS—Self-Rating Anxiety Scale, SNSB—Seoul Neuropsychological Screening Battery, CRS—chronic restraint stress, GABA—γ-aminobutyric acid, NE—norepinephrine, SCFAs—short-chain fatty acids, 5-HT—5-hydroxytryptamine/serotonin, BDNF—brain-derived neurotrophic factor, 5-HIAA—5-hydroxyindoleacetic acid, DA—dopamine, Trp—tryptophan, CORT—corticosterone, F—cortisol, DHEA—dehydroepiandrosterone, ACH—acetylcholine, EPI—epinephrine, NA—noradrenaline, His—histidine, Val—valine, Glu—glutamic acid, ADE—adrenaline, DOPAC—3,4-dihydroxyphenylacetic acid, HVA—homovalinic acid, 3-MT—3-methoxytyramine, LPS—lipopolysaccharide, IAA—indoleacetic acid, GA—glycolic acid, Gly—glycine, Xyl—xylulose, 4-HPPA—4-hydroxyphenylpropionic acid, 4-HPAA—4-hydroxyphenylacetic acid, CA—caffeic acid, LCACs—long-chain acylcarnitines, NATs—*N*-acyl taurines, OxPC—oxidized glycerophosphocholine, SMs—sphingomyelins, KA—kynurenic acid, I3PA—indole-3-propionic acid, SDStress—social defeat stress, DCS—diarrhea of college students, RPDBCT—Randomized, Placebo-Controlled, Double-Blind Clinical Trial, RCEAS—Randomized Controlled, Experimental Animal Study, EIS—Experimental Intervention Study, EPAS—Experimental Animal Study (preclinical), ECAS—Experimental Controlled Animal Study, EAS—Experimental Animal Study, RCT—Randomized Controlled Trial, RDBPCCS—Randomized, Double-Blind, Placebo-Controlled, Cross-Over Study, RDBPCT—Randomized, Double-Blind, Placebo-Controlled Trial, HBCCS-AE—Hospital-Based Cohort Study with Animal Experiments, TBRPCPS—Triple-blinded, Randomized, Placebo-Controlled Pilot Study.

## Data Availability

Not applicable.

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
