# Peer review of "Psychobiotics in Depression: Sources, Metabolites, and Treatment—A Systematic Review"

_nutrients, 2025, doi:10.3390/nu17132139_

Round 1
Reviewer 1 Report
Comments and Suggestions for Authors
Dear Authors,
I found your manuscript to be well-structured and engaging. I believe the manuscript has the potential to become a highly cited article.
I have a few minor remarks:
-
Please remove the second point before the "Protocols" section (line 132).
-
In section "2.4. Selection process," consider including some quantitative data to support your description.
-
Regarding Figure 2: I recommend removing it, as it appears unclear and does not effectively convey the data.
-
Table 1: To improve readability, please consider consolidating the results into a single column.
-
Additionally, it would be helpful to include a figure illustrating the potential mechanisms of action of probiotics on mental health.
Author Response
Comments 1: Please remove the second point before the "Protocols" section (line 132).
Response 1: Has been corrected.
Comments 2: In section "2.4. Selection process," consider including some quantitative data to support your description.
Response 2: In response to the comment regarding the inclusion of quantitative data in section 2.4 (Selection Process), we would like to clarify that detailed numerical data concerning the publication selection process have been presented in section 3.1 (Summary of studies) (line 200, p.5) in accordance with PRISMA 2020 guidelines. The methodological section focuses on describing the procedures and selection criteria, while specific numbers (369 records from three databases, 270 records screened, 114 records assessed for eligibility, 45 records included in the review) have been placed in the results section. This separation ensures transparency of the work's structure and facilitates the reader's ability to follow the flow of information from methodology to concrete results of the selection process.
Comments 3: Regarding Figure 2: I recommend removing it, as it appears unclear and does not effectively convey the data.
Response 3: We thank you for this important comment. In response to the comment regarding Figure 2, we would like to emphasize that the figure has been significantly improved in terms of visual quality and readability to better convey the presented data (line 280, p.7). Moreover, the keyword co-occurrence analysis using VOSviewer constitutes a key methodological element of this systematic review, enabling the identification of knowledge gaps that directly influenced the direction and scope of the conducted literature analysis. Additionally, a detailed characterization of the centrally positioned yellow cluster, which was not previously present in the manuscript, has been added to the main text. The supplemented description presents how the yellow cluster functions as an integrative node connecting different research areas and identifies key research gaps in the standardization of psychobiotic therapies and the translation of preclinical results to clinical populations. This figure not only visualizes the thematic structure of the literature but also provides empirical foundations for formulating recommendations for future research, making it an indispensable element of the presented work (lines 222-229, p.5).
Comments 4: Table 1: To improve readability, please consider consolidating the results into a single column.
Response 4: In response to the suggestion regarding the structure of Table 1, we have made modifications by removing the "Primary outcomes" column and retaining only secondary outcomes, which indeed simplifies the table structure (line 286, p. 8). At the same time, we would like to explain that the current division of the remaining outcomes into separate columns (behavioral outcomes, metabolites/neurotransmitters, mechanisms of action, health benefits) has significant methodological and analytical importance. This separation allows for clear differentiation of different types of study endpoints, which is crucial for proper interpretation of results. The separate presentation of metabolites and neurotransmitters as well as mechanisms of action enables identification of key biomarkers of psychobiotic action, which constitutes valuable information for future research. We believe that further merging of all outcomes into one column could significantly impair readability and analysis of different categories of endpoints.
Comments 5: Additionally, it would be helpful to include a figure illustrating the potential mechanisms of action of probiotics on mental health.
Response 5: In response to the suggestion regarding the inclusion of an additional figure illustrating the mechanisms of probiotic action on mental health, we would like to explain that a graphical abstract has been attached to the article, which synthetically presents the key elements of the discussed topic. Due to the considerable size of this work, adding further illustrations would exceed the accepted scope limits of the publication. At the same time, in response to this valuable comment, we would like to emphasize that the discussion section thoroughly addresses the topic of probiotic mechanisms of action on mental health, which allows for an in-depth examination of this issue in textual form.
Reviewer 2 Report
Comments and Suggestions for Authors
Dear colleagues,
Thank you for the opportunity to review the article "Psychobiotics in Major Depressive Disorder: sources, metabolites, and treatment – a systematic review."
The first comment is that it is an interesting topic that captures the reader's attention.
Below you will find my detailed comments:
Although the article is linguistically correct, we kindly ask the authors to perform a language check and correct some minor issues
Title
Attractive. However, the title mentions sources, metabolites, and treatment, while the abstract talks about sources, mechanisms, and future directions. It would be good to harmonise the terminology between the abstract and the title.
Abstract
Lines 29-30: "...real-world applications in food-based interventions..." – unclear statement that should be clarified.
Keywords
Lines 31-32: The terms “depression, anxiety, insomnia, cognitive disorders” – I ask the authors to explain the reason for the choice of these keywords. In particular, "major depressive disorder" would be a sufficient keyword to replace the above examples. Considering the diverse manifestations of this disorder, the authors have chosen only a few of them. However, if they have included, for example, insomnia, why have they not included hypersomnia, which may also be one of the manifestations of MDD? I suggest that they address this comment or change the keywords.
Introduction
Just a comment for the authors, no need for a reply. Although there is no word limit in this journal, the introduction is too long. In the future, I advise authors to focus on the main objectives of their work and briefly explain the topic. Although everything written is fine, an introduction that is too long discourages the reader from continuing to read the text.
Materials and methods
Line 132: “2.1. . Protocol” – I ask the authors to delete the period before the word protocol.
2.2. Eligibility criteria
Lines 139-140: “They had to involve…”The authors mention the inclusion of studies with “mental illness (e.g., depression, anxiety, stress-related disorders).” I suggest that the authors explain these statements in more detail. In particular, this expansion of diagnoses beyond MDD in the selection of studies may lead to confusion.
Lines 140-141: “…healthy individuals in whom effects on mental well-being were assessed…” The inclusion of studies on healthy individuals in whom the effects on psychological well-being were investigated may also confuse. I would therefore advise the authors to explain in more detail what this sentence refers to.
2.3. Information sources and search strategy
Lines 149-150: "...To ensure the currency of the analyzed…” — Considering that psychobiotics is a relatively new field of research, there are certainly high-quality studies that have been published earlier (I will not list these studies here to remain objective). Therefore, I suggest that the authors briefly address this fact in the text and further explain their choice of the specific period.
2.4. Selection process
Lines 166-167:
“…adequacy to the adopted criteria..." — The authors probably mean “compliance with the inclusion criteria", which is a simpler and clearer expression, although the wording used is not incorrect.
2.5. Data extraction
Lines 181-182:
“Discrepancies were resolved by consensus with a third author.” — This wording is vague. I suggest that the authors briefly clarify how disagreements were identified, documented, or resolved, and whether there was a predefined protocol for this process.
2.6. Risk of bias in individual studies
Line 184: “…Tree authors” – This is a minor grammatical error; it should read “Three authors.” I ask the authors to correct this.
We also ask the authors to mention that three authors have been trained in the use of the ROB 2 tool to ensure consistent and reliable assessments.
Results
3.1. Summary of studies
Line 249: "...Cristensenella..." – The correct term is Christensenella. I ask the authors to correct the spelling and ensure consistency with the graphical representation (Figure 3A).
Line 254: "...psychobiotic candidate..." – This expression is a bit clumsy. I ask the authors not to use it or to clarify its meaning.
Line 256: "...37,8% each..." – Please correct the comma to a full stop (i.e. 37.8%) to conform to English decimal notation.
Discussion
4.1. Main Findings - Unnecessary subheading; I advise the authors to remove it, as it does not add significant value.
4.2. Limitations of the evidence included in the review – I assume the authors mean the limitations of the studies included in the review?
4.3. Implications of the results for practice and policy
Although this section is well written, the authors have turned it back into a discussion. I suggest that the authors shorten this entire section and state the importance of this research clearly and without excessive explanation.
Lines 471-474:
“The results..." — The authors emphasise the importance of their findings, but elsewhere in the "Discussion" and "Limitations" sections, they acknowledge numerous shortcomings of previous studies. Therefore, this opening sentence contradicts earlier statements, and it would be advisable to formulate it more softly and diplomatically.
References
I advise authors to standardise the way references are written and adapt them to the journal's guidelines.
It is necessary to standardise the citation style.
Here are just a few examples, but I ask authors to check all references:
In reference 1, the authors mention “2022, Vol. 13, Page 646 2022, 13, 646” twice.
In references 2 and 4, they mention the year of publication and then the DOI, while in references 3 and 5, they mention the year of publication, the volume, the article number, and only then the DOI.
In reference 10, the year of publication and the volume are mentioned, but neither the page numbers nor the article number are, then the DOI is added.
Comments on the Quality of English LanguageAlthough the article is linguistically correct, we kindly ask the authors to perform a language check and correct some minor issues
Author Response
Comments 1: Title. Attractive. However, the title mentions sources, metabolites, and treatment, while the abstract talks about sources, mechanisms, and future directions. It would be good to harmonise the terminology between the abstract and the title.
Response 1: We thank you for this important comment regarding the consistency of terminology between the title and abstract. In response to this suggestion, we have changed the title to "Psychobiotics in Depression: sources, metabolites, and treatment - a systematic review" to better reflect the issues addressed in the article. Additionally, the sentence in the abstract has been modified to "Psychobiotics show significant potential as adjunctive and therapeutic agents in depressive disorders through modulation of the gut-brain axis," which ensures greater terminological consistency and better reflects the scope of the conducted analysis. (lines 27-28, p.1)
In addition, the paragraph (lines 22-27) in the Abstract has been reworded to also draw attention to the metabolites of psychobiotic strains.
Comments 2: Abstract. Lines 29-30: "...real-world applications in food-based interventions..." – unclear statement that should be clarified.
Response 2: We thank you for the comment regarding the unclear formulation in lines 29-30 referring to "real-world applications in food-based interventions." We agree that this term was imprecise and could be misleading. In response to this suggestion, the formulation has been removed from the text.
Comments 3: Keywords. Lines 31-32: The terms “depression, anxiety, insomnia, cognitive disorders” – I ask the authors to explain the reason for the choice of these keywords. In particular, "major depressive disorder" would be a sufficient keyword to replace the above examples. Considering the diverse manifestations of this disorder, the authors have chosen only a few of them. However, if they have included, for example, insomnia, why have they not included hypersomnia, which may also be one of the manifestations of MDD? I suggest that they address this comment or change the keywords.
Response 3: We thank you for this pertinent comment regarding the choice of keywords in lines 31-32. We completely agree with the presented argumentation - indeed, listing only selected symptoms such as "depression, anxiety, insomnia, cognitive disorders" was inconsistent and could be misleading. In response to this comment, we have removed the mentioned terms from the keywords and retained the remaining, more precise terms that better reflect the scope of our review. We thank you for pointing out this terminological inconsistency (lines 29-30, p.1).
Comments 4: Introduction. Just a comment for the authors, no need for a reply. Although there is no word limit in this journal, the introduction is too long. In the future, I advise authors to focus on the main objectives of their work and briefly explain the topic. Although everything written is fine, an introduction that is too long discourages the reader from continuing to read the text.
Response 4: Nevertheless, we have taken advantage of this valuable comment. In response to this suggestion, the introduction has been shortened and at the same time adapted to the new title of the article (lines 32-117 p. 1-3).
Comments 5: Materials and methods. Line 132: “2.1. . Protocol” – I ask the authors to delete the period before the word protocol.
Response 5: Has been corrected.
Comments 6: 2.2. Eligibility criteria. Lines 139-140: “They had to involve…”The authors mention the inclusion of studies with “mental illness (e.g., depression, anxiety, stress-related disorders).” I suggest that the authors explain these statements in more detail. In particular, this expansion of diagnoses beyond MDD in the selection of studies may lead to confusion.
Response 6: We thank you for this important comment regarding the scope of included diagnoses in lines 139-140. In response to this suggestion, the title, abstract, and introduction have been modified to better reflect the broader topic addressed throughout the work. We deliberately focused not only on depression (MDD) but also on other stress-related psychiatric disorders to more thoroughly demonstrate the scale of the problem and the complexity of psychobiotic action in the context of various psychopathological states. Additionally, the "Eligibility criteria" paragraph has been corrected to increase the precision of inclusion criteria. (line 124, p.3) This approach allowed for a more comprehensive analysis of available scientific evidence and a better understanding of the spectrum of psychobiotic applications in psychiatry. We thank you for drawing attention to the necessity of precisely defining the scope of studied populations.
Comments 7: Lines 140-141: “…healthy individuals in whom effects on mental well-being were assessed…” The inclusion of studies on healthy individuals in whom the effects on psychological well-being were investigated may also confuse. I would therefore advise the authors to explain in more detail what this sentence refers to.
Response 7: We thank you very much for this pertinent comment regarding the ambiguity of the formulation in lines 140-141. Indeed, the term "healthy individuals in whom the impact on mental well-being was assessed" could introduce confusion regarding the characteristics of the studied population. In response to this suggestion, the formulation has been changed to "undiagnosed individuals exhibiting symptoms of stress-related psychiatric disorders," which precisely defines the group of individuals without formal diagnosis but showing symptoms of stress-related disorders. (lines 128-129 p. 3)
Comments 8: 2.3. Information sources and search strategy. Lines 149-150: "...To ensure the currency of the analyzed…” — Considering that psychobiotics is a relatively new field of research, there are certainly high-quality studies that have been published earlier (I will not list these studies here to remain objective). Therefore, I suggest that the authors briefly address this fact in the text and further explain their choice of the specific period.
Response 8: Thank you for your advice regarding its important correction. After your comment we modified the body text of chapter 2.3 by adding sentence: It is important to note that psychobiotics is a relatively new area of research, with several high-quality studies and narrative reviews on the topic published over the past decade [29-31]. However, to ensure the currency of the analyzed data, a five-year publication timeframe 2020-2024 was adopted. (lines 138-141 p. 3)
Comments 9: 2.4. Selection process. Lines 166-167: “…adequacy to the adopted criteria..." — The authors probably mean “compliance with the inclusion criteria", which is a simpler and clearer expression, although the wording used is not incorrect.
Response 9: Has been corrected.
Comments 10: 2.5. Data extraction. Lines 181-182: “Discrepancies were resolved by consensus with a third author.” — This wording is vague. I suggest that the authors briefly clarify how disagreements were identified, documented, or resolved, and whether there was a predefined protocol for this process.
Response 10: In response to the comment regarding the process of resolving discrepancies, the sentence has been expanded with methodological details according to systematic review guidelines. The corrected formulation reads: "Three authors independently extracted data from each accepted study using the Rayyan® platform (as shown in Table 1). Discrepancies between authors were automatically identified through the platform's blind review system (blind on/off function), systematically documented, and resolved by consensus with an independent third author following pre-defined protocols for systematic review methodology" (lines 172-176, p. 4). The Rayyan platform enables independent assessment by hiding other reviewers' decisions during the article selection process, ensuring objectivity and eliminating potential bias in decisions about inclusion or exclusion of publications. We thank you for drawing attention to the necessity of clarifying this crucial methodological aspect.
Comments 11: 2.6. Risk of bias in individual studies Line 184: “…Tree authors” – This is a minor grammatical error; it should read “Three authors.” I ask the authors to correct this.
Response 11: Has been corrected.
Comments 12: We also ask the authors to mention that three authors have been trained in the use of the ROB 2 tool to ensure consistent and reliable assessments.
Response 12: We thank you for this important comment regarding the methodology of systematic bias risk assessment. In response to this suggestion, the following sentence has been added: "Three authors were trained in the use of the Revised Cochrane Risk-of-Bias Tool for Randomized Trials (ROB 2) according to official training materials to ensure consistent and reliable assessments." (lines 178-180, p.4). The training was conducted based on the official guide available at https://www.riskofbias.info/welcome/rob-2-0-tool/current-version-of-rob-2.
Comments 13: Results. 3.1. Summary of studies. Line 249: "...Cristensenella..." – The correct term is Christensenella. I ask the authors to correct the spelling and ensure consistency with the graphical representation (Figure 3A).
Response 13: Has been corrected.
Comments 14: Line 254: "...psychobiotic candidate..." – This expression is a bit clumsy. I ask the authors not to use it or to clarify its meaning.
Response 14: We thank you very much for the comment regarding the expression "psychobiotic candidates" in line 254. Indeed, this formulation could be unclear. In response to this suggestion, the sentence has been corrected to: "This distribution reflects the diverse origin of psychobiotic strains and indicates potential areas for more detailed reporting in future studies." (line 262, p.6)
Comments 15: Line 256: "...37,8% each..." – Please correct the comma to a full stop (i.e. 37.8%) to conform to English decimal notation.
Response 15: Has been corrected.
Comments 16: Discussion. 4.1. Main Findings - Unnecessary subheading; I advise the authors to remove it, as it does not add significant value.
Response 16: Has been corrected.
Comments 17: 4.2. Limitations of the evidence included in the review – I assume the authors mean the limitations of the studies included in the review?
Response 17: Has been corrected.
Comments 18: 4.3. Implications of the results for practice and policy. Although this section is well written, the authors have turned it back into a discussion. I suggest that the authors shorten this entire section and state the importance of this research clearly and without excessive explanation.
Response 18: This section has been significantly shortened in line with the reviewer's suggestion (lines 496-510, p. 22).
Comments 19: Lines 471-474: “The results..." — The authors emphasise the importance of their findings, but elsewhere in the "Discussion" and "Limitations" sections, they acknowledge numerous shortcomings of previous studies. Therefore, this opening sentence contradicts earlier statements, and it would be advisable to formulate it more softly and diplomatically.
Response 19: We thank you for this pertinent comment regarding the consistency of the message in the results and discussion sections. Indeed, the opening sentence of the results section was formulated too categorically in the context of limitations identified in the analyzed studies, which we discuss in detail in the later part of the work. In response to the reviewers' suggestions, the entire results section has been shortened and modified toward a more balanced and diplomatic formulation that better reflects the complexity of the current state of knowledge about psychobiotics (lines 481-495 p. 22).
Comments 20: References. I advise authors to standardise the way references are written and adapt them to the journal's guidelines. It is necessary to standardise the citation style.
Response 20: Has been corrected.
Reviewer 3 Report
Comments and Suggestions for Authors
This study is well prepared and appropriate for publication.
However, the following points should be addressed before publication.
This review study was conducted with very recent papers of limited period of 2020-2024. The following similar three reviews have been published before 2020. So, what is new in this review compared with the previous studies before 2020. The contents provided by previous reviews are similar to those of this study. The most important originality and impact in this review is unclear, although extensive information was provided.
1. Josipa Vlainić Vlainić et al. Probiotics as an Adjuvant Therapy in Major Depressive Disorder. Curr Neuropharmacol. 2016; 14(8):952-958.
2. Adiel C Rios et al. Microbiota abnormalities and the therapeutic potential of probiotics in the treatment of mood disorders. Rev Neurosci. 2017; 28(7):739-749.
3. Ibrahim Nadeem et al. Effect of probiotic interventions on depressive symptoms: A narrative review evaluating systematic reviews. Psychiatry Clin Neurosci. 2019; 73(4):154-162.
In Table 1, information for the administration periods of probiotics are necessary.
In Table 1 (p3), “Betaine”. Is this correct?
Author Response
Comments 1: This review study was conducted with very recent papers of limited period of 2020-2024. The following similar three reviews have been published before 2020. So, what is new in this review compared with the previous studies before 2020. The contents provided by previous reviews are similar to those of this study. The most important originality and impact in this review is unclear, although extensive information was provided. 1. Josipa Vlainić Vlainić et al. Probiotics as an Adjuvant Therapy in Major Depressive Disorder. Curr Neuropharmacol. 2016; 14(8):952-958. 2. Adiel C Rios et al. Microbiota abnormalities and the therapeutic potential of probiotics in the treatment of mood disorders. Rev Neurosci. 2017; 28(7):739-749. 3. Ibrahim Nadeem et al. Effect of probiotic interventions on depressive symptoms: A narrative review evaluating systematic reviews. Psychiatry Clin Neurosci. 2019; 73(4):154-162.
Response 1: We thank the reviewer for pointing out valuable publications concerning psychobiotics. In response to this comment, we have modified the text of section 2.3, adding the sentence: "It is important to note that psychobiotics is a relatively new area of research, with several high-quality studies and narrative reviews on the topic published over the past decade [29-31]. However, to ensure the currency of the analyzed data, a five-year publication timeframe 2020-2024 was adopted." (lines 138-141, p. 3)
The indicated publications constitute a valuable contribution to the field of psychobiotics and we truly appreciate them. However, they differ from our work in methodological and temporal terms. The cited articles (2016-2019) are narrative reviews focusing on general assessment of probiotic potential and discussion of theoretical mechanisms. Our work constitutes a systematic review according to PRISMA guidelines, concentrating on the most recent empirical studies from 2020-2024. This temporal scope allowed us to fill the knowledge gap with the most current data concerning strain characteristics, metabolites, and mechanisms of psychobiotic action based on confirmed results from clinical and preclinical studies. Additionally, our methodological approach enabled systematic assessment of evidence quality and identification of areas requiring further research, which constitutes a valuable supplement to the existing literature.
Comments 2: In Table 1, information for the administration periods of probiotics are necessary.
Response 2: In response to the comment regarding Table 1, information concerning probiotic administration periods has been supplemented, as well as bacterial/strain quantities according to the reviewers' suggestion, and column headers have been modified to be more comprehensible and readable. (line 286, p. 8)
Comments 3: In Table 1 (p8), “Betaine”. Is this correct?
Response 3: In response to the comment regarding Table 1 (p. 8) and the term "Betaine," we thank you very much for this valid observation. After re-analysis, we admit that we made an error - betaine should not be included in this table in the context of bacterial metabolites. Consequently, the term "betaine" has been removed from Table 1 (p. 8). We thank you for your attention and thoroughness in the review.
Reviewer 4 Report
Comments and Suggestions for Authors
This systematic review aimed to determine the effects of psychobiotic microorganisms on mental health outcomes, with particular focus on their sources, metabolites, and therapeutic potential for depression. The review followed PRISMA guidelines and was conducted using publications from 2020-2024 found in Web of Science, Scopus and PubMed databases. The gut microbiome plays a key role in regulating mood, so a diet rich in pro- and prebiotics is an important part of preventing mental disorders.
Some suggestions:
1.Abstract, lines 25-27, you wrote “Psychobiotics show significant potential as adjunctive and preventive agents in depressive disorders…The title of the article is “ Psychobiotics in Major Depressive Disorder”. If they are preventive agents the title is not suitable. In introduction you wrote also about major depressive disorders.
2. Lines 36-37: you wrote that major depressive disorders affects over 300 million people worldwide. For which year is this statistic valid? Please add.
- Line 72: add please some details about the “inactivated microorganisms with similar mental health benefits”.
- Point 2.2. Eligibility criteria, lines 139-140 – you followed also participants diagnosed with other mental health conditions such as anxiety or stress-related disorders not only with depression?
In my opinion the inclusion/exclusion criteria are not clearly defined. Please improve them.
5.Lines 222-223, you wrote “Terms related to animal models, such as rats and expressions, were prominently associated within this cluster.” What do you mean by rats and expressions? Please clarify.
6.Please try to improve the quality of Figure 2.
- Concerning the studies presented 1n Table 1: What amount of species/strains were administrated and how long? You didn’t add for all the studies in what form they are administered. Please complete.
Also, In table 1 the studies on animals are mixed with the studies on humans. Please separate them. You can present in a table the studies on animals and in an other table the studies on humans.
- Table 1, concerning the Lan et al., 2023 [40] study. You wrote as primary outcome – Daidzein. How is this formed?
9.Discussions – give please more details about the mechanisms of action of the bacterial species in major depressive disorder.
Author Response
Comments 1: Abstract, lines 25-27, you wrote “Psychobiotics show significant potential as adjunctive and preventive agents in depressive disorders…The title of the article is “ Psychobiotics in Major Depressive Disorder”. If they are preventive agents the title is not suitable. In introduction you wrote also about major depressive disorders.
Response 1: In response to this comment, we thank you for drawing attention to this terminological inconsistency. Indeed, the use of the term "preventive" in the context of depressive disorders was imprecise. Our study focuses on analyzing the effectiveness of psychobiotics in treating existing depressive disorders, not on their preventive role. Therefore, a terminological correction will be made in the abstract - the word "preventive" will be replaced with the term "therapeutic" to maintain consistency with the article title and actual content of the work. The sentence will read: "Psychobiotics show significant potential as adjunctive and therapeutic agents in depressive disorders," which better reflects the purpose and scope of the conducted systematic review (line 27, p.1).
Comments 2: Lines 36-37: you wrote that major depressive disorders affects over 300 million people worldwide. For which year is this statistic valid? Please add.
Response 2: In response to the comment regarding statistics on depressive disorders, we thank you for drawing attention to the need for precision in epidemiological data. The statistic has been updated according to the latest WHO data from 2023, which indicates that depressive disorders affect approximately 280 million people worldwide. A temporal reference for this statistic has been added to the text, which increases its credibility and currency. The corrected sentence now reads: "Recent 2023 data indicates that approximately 280 million people are affected by depression worldwide, making it the most prevalent psychiatric condition and the leading cause of disability globally" (line 35, p.1). In response to reviewers' suggestions, changes have also been introduced in the introduction and publication title, which now more precisely reflect the scope and nature of the conducted analysis.
Comments 3: Line 72: add please some details about the “inactivated microorganisms with similar mental health benefits”.
Response 3: In response to the suggestion regarding supplementing information about inactivated microorganisms, the text has been enriched with a description of postbiotics by adding the sentences: "Inactivated microbial cells, also known as postbiotics, include metabolites, inactivated cells, and other molecules that support the development of psychobiotic strains in the gut. The use of inactivated microorganisms has several advantages over live organisms, including the lack of risk of infection in susceptible individuals and ease of use in terms of storage and administration" (lines 70-74, p. 2).
Comments 4: Point 2.2. Eligibility criteria, lines 139-140 – you followed also participants diagnosed with other mental health conditions such as anxiety or stress-related disorders not only with depression? In my opinion the inclusion/exclusion criteria are not clearly defined. Please improve them.
Response 4: We thank you very much for these significant suggestions regarding the scope of studied disorders. In response to this valuable comment, both the title and introduction have been modified to better reflect the broader scope of addressed issues. The title now reads: "Psychobiotics in Depression: sources, metabolites, and treatment - a systematic review." This modification precisely reflects the fact that our review included not only studies concerning depression but also participants with anxiety disorders and other stress-related disorders, which has been appropriately reflected in the eligibility criteria. (line 124, p. 3)
Comments 5: Lines 222-223, you wrote “Terms related to animal models, such as rats and expressions, were prominently associated within this cluster.” What do you mean by rats and expressions? Please clarify.
Response 5: In response to the comment regarding lines 222-223, we thank you for drawing attention to this imprecision in formulation. Indeed, the expression "rats and expressions" was imprecise and could be misleading. It was intended to refer to terms related to animal models (such as rats) and neurobiological processes (such as expression of signaling molecules). The sentence has been corrected to: "Terms related to animal models (such as rats) and neurobiological processes (such as expression of neural signaling molecules) were prominently associated within this cluster." We thank you for the comment that allowed for increased precision of the message. (lines 216-218, p. 5)
Comments 6: Please try to improve the quality of Figure 2.
Response 6: In response to the comment regarding the quality of Figure 2, the visual quality of the figure has been significantly improved to increase readability and clarity of the presented data.
Comments 7: Concerning the studies presented 1n Table 1: What amount of species/strains were administrated and how long? You didn’t add for all the studies in what form they are administered. Please complete. Also, In table 1 the studies on animals are mixed with the studies on humans. Please separate them. You can present in a table the studies on animals and in an other table the studies on humans.
Response 7: In response to the comment regarding Table 1, information concerning probiotic administration periods and bacterial/strain quantities has been supplemented according to the reviewer's suggestion. Additionally, for better clarity and data organization, the table has been divided into two separate parts: one presenting human studies and another presenting animal studies, and column headers have been improved to be more comprehensible. (line 286, p.8)
Comments 8: Table 1, concerning the Lan et al., 2023 [40] study. You wrote as primary outcome – Daidzein. How is this formed?
Response 8: In response to the comment regarding the Lan et al., 2023 [40] study, we thank you for drawing attention to this classification. A correction has been made by moving daidzein from the "Bacterial Metabolites/Neurotransmitters" column to the "Mechanism" column, where it has been described as "Stress marker reduction by daidzein." Daidzein is a phytoestrogenic isoflavonoid, not a bacterial metabolite, therefore its original placement in the metabolites column was incorrect. The corrected version better reflects the actual role of daidzein as a factor modulating stress markers in the probiotic's mechanism of action.
Comments 9: Discussions – give please more details about the mechanisms of action of the bacterial species in major depressive disorder.
Response 9: Thank you for this suggestion. Following your comment, we thoroughly reanalyzed the content of the discussion in light of the objectives of our systematic review and the current state of knowledge. Ultimately, we decided not to include further details regarding the mechanisms of action of specific bacterial species. Firstly, the recognized mechanisms underlying the antidepressant effects of the bacterial species discussed are already presented in Table 1, further elaborated in Section 4.1 of the Discussion, and as well as in the Conclusions. Secondly, despite the growing body of knowledge on the gut microbiome, the molecular mechanisms through which it influences central and systemic levels of 5-HT, NE, DA, GABA, BDNF, and inflammatory processes remain poorly understood. Therefore, including speculative interpretations in this area would go beyond the scope of the present review. Finally, we believe that readers interested in the detailed mechanisms of action will find valuable guidance in the references provided throughout the manuscript [29-31].
Round 2
Reviewer 4 Report
Comments and Suggestions for Authors
The manuscript has been significantly improved. The authors addressed all the required issues.
Best wishes!